# MoVQ: Modulating Quantized Vectors for High-Fidelity Image Generation

**Chuanxia Zheng**
Monash University
chuanxiazheng@gmail.com

**Long Tung Vuong**
VinAI
longvt94@gmail.com

**Jianfei Cai**
Monash University
Jianfei.Cai@monash.edu

**Dinh Phung**
Monash University
dinh.phung@monash.edu

## Abstract

Although two-stage Vector Quantized (VQ) generative models allow for synthesizing high-fidelity and high-resolution images, their quantization operator encodes similar patches within an image into the same index, resulting in a repeated artifact for similar adjacent regions using existing decoder architectures. To address this issue, we propose to incorporate the spatially conditional normalization to modulate the quantized vectors so as to insert spatially variant information to the embedded index maps, encouraging the decoder to generate more photorealistic images. Moreover, we use multichannel quantization to increase the recombination capability of the discrete codes without increasing the cost of model and codebook. Additionally, to generate discrete tokens at the second stage, we adopt a Masked Generative Image Transformer (MaskGIT) to learn an underlying prior distribution in the compressed latent space, which is much faster than the conventional autoregressive model. Experiments on two benchmark datasets demonstrate that our proposed modulated VQGAN is able to greatly improve the reconstructed image quality as well as provide high-fidelity image generation.

## 1 Introduction

The vision community has rapidly improved image synthesis results on quality, diversity and resolution over a short period of time. In particular, many powerful baseline frameworks have been introduced, such as Generative Adversarial Networks (GAN) [13], Variational AutoEncoder (VAE) [23], Flow-based Generators [8] and Diffusion Probabilistic Models (DPM) [35]. These methods are conceptually intuitive, leading to an explosion of image synthesis works that push the boundaries of image generation quality and diversity [14, 1, 22, 20, 21, 37, 4].

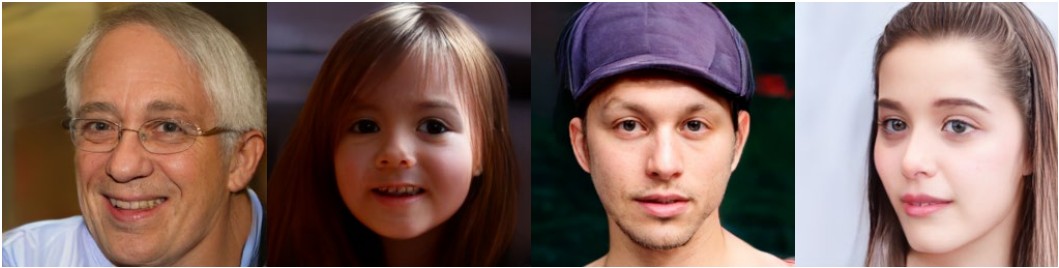

Figure 1: 256×256 image samples generated by the proposed MoVQ, with model trained on FFHQ.

Among these existing image synthesis frameworks, Vector Quantized-Variational AutoEncoder (VQ-VAE) [39] is a very popular nova, which learns a compressed discrete representation for each image

36th Conference on Neural Information Processing Systems (NeurIPS 2022).

at the first stage and subsequently learns the underlying prior distribution in the discrete latent space at the second stage. Unlike conventional GANs that pursue a fragile balance in a minimax game, resulting in unstable training, the optimization target of VQ-VAE is a definite negative log-likelihood (NLL) objective of the training data. By maximizing the log-likelihood over *all* training examples, theoretically VQ-VAE is able to cover all modes of the data, and thus bypass the "mode collapse" issue in GAN. On the other hand, compared with VAE [23], VQ-VAE maps an image into a palette of latent discrete codes in higher resolution with spatial structure information and learns the composition of the codes from the data itself, which overcomes the long-dragged image quality issue of VAE in image synthesis. These advantages of VQ-VAE have led to remarkable image synthesis results as evident by its recent extensions, such as VQ-VAE-2 [31], DALL-E [30], VQGAN [11], ImageBART [10], LDMs [32], VIT-VQGAN [45], RQ-VAE [24], MaskGIT [2] and DALL-E-2 [29].

Despite the great performance, the VQ-VAE or VQGAN pipeline also has its shortcomings. For example, its second stage is typically modelled as a sequence generation process in an autoregressive way [39, 11], i.e. generating each discrete latent code one by one at different spatial locations, which is very time-consuming for inference. MaskGIT [2] nicely addresses this issue by predicting multiple tokens based on the prediction confidence at each step, which greatly reduces the number of steps needed in the autoregressive token generator.

In contrast, in this paper we focus on improving the stage-1 learning of the VQ-based image synthesis pipeline. Specifically, we notice another inherent shortcoming of the VQ-based methods, i.e. they often generate repeated artifact patterns in image synthesis (see Fig. 3), which is because the quantization operator drops away some variances and embeds similar patches into the same quantization index. To address this, motivated by the spatially conditional normalization and position embedding literature [17, 27, 20, 41], we re-design the decoder architecture in a way that utilizes a spatially conditional normalization layer to modulate the activations using the quantized vectors, which essentially adds spatially variant information to the discrete representation. Moreover, we leverage a multichannel representation technique, introduced in [47] for image completion, to keep the codebook size manageable while with sufficient representation capability for image generation. Lastly, with a better quantizer, we follow MaskGIT [2] to speed up the prior distribution training over the compact discrete representations in the second stage [45].

In summary, our main contributions are as follows:

- We point out an inherent shortcoming of the VQ-based image synthesis pipeline: having repeated artifacts in semantically similar nearby regions, and solve it by introducing a spatially conditional normalization to provide spatially variant information for different locations.
- With the help of our proposed spatially conditional normalization, we further incorporate the multichannel representation that dramatically improves the reconstructed image quality using the same encoder-decoder layers as in VQGAN.
- When sampled from a learned prior on all directions for multi-locations and multi-channels, experimental results on two benchmark datasets show that our synthesized samples hold not only high quality but also large diversity.

## 2 Related Work

**VQ-based image synthesis.**   Our model is derived from VQGAN [11], which is built upon the two-stage VQ-VAE [39]: first, a quantizer with an encoder-decoder architecture is trained to embed images into compact sequences using discrete tokens from a learned codebook, and then a prior network is learned to model the underlying distribution in the discrete space. Unlike the minimax game in popular GAN models [13, 1, 20, 21], the VQ-based generator is trained by optimizing negative log-likelihood over all examples in the training set, leading to a stable training and bypassing the "mode collapse" issue. Driven by these advantages, many image synthesis models follow the two-stage paradigm, such as image generation [31, 45, 2, 24, 16], image-to-image translation [11, 10, 32], text-to-image synthesis [30, 29, 10, 7], conditional video generation [28, 42, 44], and image completion [11, 10, 47]. Apart from VQGAN, the most related works also include ViT-VQGAN [45] and RQ-VAE [24] that aim to train a better quantizer in the first stage. Compared to them, our model is simple and efficient, yet effective to improve the image quality, without adding the computational cost on higher resolution representations with much larger model [45] or more stages of recursive quantization [24].

**Spatially conditional normalization,** also called adaptive instance normalization, has been widely used in image synthesis task [9, 17, 18, 27, 20, 41]. Among them, the spatially conditional input contains various variants, such as style images [9, 17], target domain images [18], semantic maps [27], learned vectors [20] and random heatmaps [41]. Comparing our approach to these works, our spatially conditional input is the embedded discrete features, which contain automatically learned compact contents.

To the best of our knowledge, this is the first work to modulate quantized vectors and use multichannel quantization on the VQ-based image generation framework. In the following sections, we will describe the modulated quantized vectors and multichannel quantization and discuss their advantages over the concurrent models such as [32, 45] and [24] in detail.

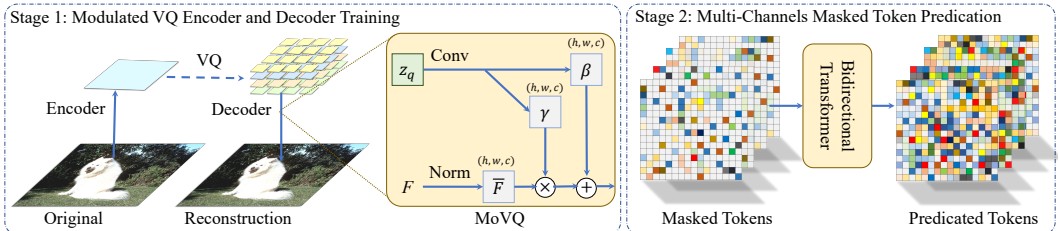

Figure 2: Left: The quantizer architecture of our proposed MoVQ. We incorporate the spatially conditional normalization layer into the decoder, where the two convolution layers predict modulation parameters $\gamma$ and $\beta$ in a point-wise way to modulate the learned discrete structure information. Right: Masked image generation. Here, a bidirectional transformer is applied to estimate the underlying prior distribution on the discrete representation with multiple channels.

## 3 Method

The proposed approach, illustrated in Fig. 2, follows a two-stage recipe that embeds images using a learnable codebook (see Fig. 2 left), and then tames a bidirectional transformer to estimate the underlying prior distribution over the discrete latent space (see Fig. 2 right). The *key observation* we found is that a better quantizer will naturally lead to better image reconstruction and generation quality [45]. Hence, our *main goal* in this work is to improve the quantization in the first stage, yet without suffering from awkward computational cost as in concurrent works [45, 24].

### 3.1 Modulating Quantized Vector

**Background.** Given an image $x \in \mathbb{R}^{H \times W \times 3}$, vanilla VQ-VAEs learn a discrete codebook to represent observations as a collection of codebook entries $z_q \in \mathbb{R}^{h \times w \times n_q}$, where $n_q$ is the dimensionality of quantized vectors in the codebook. In this way, each image can be equivalently represented as a compact sequence $\mathbf{s}$ with $h \cdot w$ indices of the codevectors $z_q$. Formally, the observed image $x$ is reconstructed by:

$$\hat{x} = \mathcal{G}_\theta(z_q) = \mathcal{G}_\theta(\mathbf{q}(\hat{z})) = \mathcal{G}_\theta(\mathbf{q}(\mathcal{E}_\psi(x))). \tag{1}$$

In particular, an encoder $\mathcal{E}_\psi(\cdot)$ first embeds an image $x$ into a continuous vector $\hat{z}$, and the quantization operator $\mathbf{q}(\cdot)$ is then conducted to transfer the continuous feature $\hat{z}$ into the discrete space by looking up the closest codebook entry $z_k$ for each spatial grid feature $\hat{z}_{ij}$ within $\hat{z}$:

$$z_q = \mathbf{q}(\hat{z}) = \underset{z_k \in \mathcal{Z}}{\arg\min} \|\hat{z}_{ij} - z_k\|, \tag{2}$$

where $\mathcal{Z} \in \mathbb{R}^{K \times n_q}$ is the codebook that consists of $K$ entries with $n_q$ dimensions. The quantized vector $z_q$ is finally transmitted to a decoder $\mathcal{G}_\theta(\cdot)$ for rebuilding the original image. The overall models and the codebook can be learned by optimizing the following objective:

$$\mathcal{L}(\mathcal{E}_\psi, \mathcal{G}_\theta, \mathcal{Z}) = \|x - \hat{x}\|_2^2 + \|\mathrm{sg}[\mathcal{E}_\psi(x)] - z_q\|_2^2 + \beta \|\mathrm{sg}[z_q] - \mathcal{E}_\psi(x)\|_2^2. \tag{3}$$

Here, sg denotes the stop-gradient operator, and $\beta$ is a hyperparameter for the third *commitment loss*. The first term is a *reconstruction loss* to estimate the error between the observed $x$ and the reconstructed $\hat{x}$. The second term is the *codebook loss* to optimize the entries in the codebook. We use the released VQGAN implementation as our baseline.

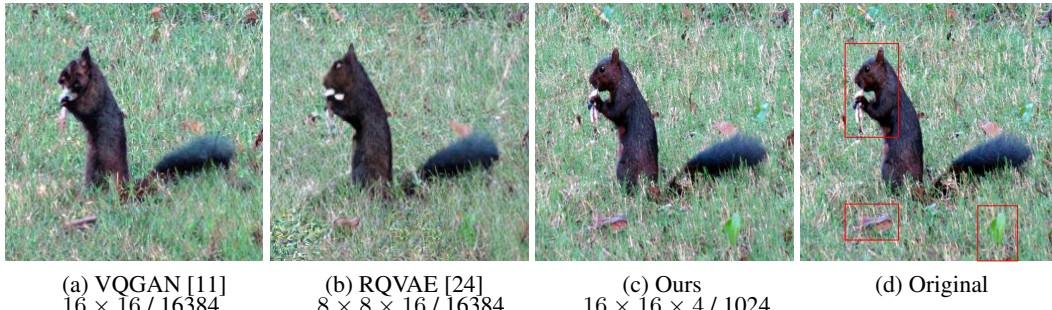

| (a) VQGAN [11] | (b) RQVAE [24] | (c) Ours | (d) Original |
|:---:|:---:|:---:|:---:|
| $16 \times 16$ / 16384 | $8 \times 8 \times 16$ / 16384 | $16 \times 16 \times 4$ / 1024 | |

Figure 3: Reconstructions from different models. The numbers denote the represented latent size and learned codebook sizes, respectively. Compared to the latest state-of-the-art RQVAE [24], our model dramatically improves the image quality in the first stage under the same compression ratio.

**Spatially conditional normalization.** The quantization operator is lossy [31], and similar patches are often embedded as the same codebook indices, resulting in a repeated artifact when they are synthesized through existing decoder architectures (see Fig. 3 (a)). As opposed to existing methods that directly feed discrete feature maps into the decoder as activations, in this work, we introduce a new spatially conditional normalization layer to propagate the embedded contents to the activations. The *key motivation* behind this is to add spatial variants to the discrete maps, such that the model can generate plausible and diverse results, even for the same quantization index in neighboring regions.

The structure of our modulated decoder is illustrated in Fig. 2 (left), with an activation $F$ normalized by a conventional normalization, and then modulated by the learned scale and bias calculated from the embedded vector. Specifically, the activation $F$ of the $i$-th layer in the decoder $\mathcal{G}_\theta$ is given by:

$$F^i = \phi_\gamma(z_q)\frac{F^{i-1} - \mu(F^{i-1})}{\sigma(F^{i-1})} + \phi_\beta(z_q), \tag{4}$$

where $F^{i-1}$ is the intermediate feature map, which can be initialized as the positional embedding [12, 40], learned constant [20], or Fourier features [36, 19] for $F^0$ in the decoder. $\mu(\cdot)$ and $\sigma(\cdot)$ respectively denote the functions for calculating the mean and standard deviation of the activation. There are many choices of the normalization. Following the baseline VQGAN [11], here we directly use the Group Normalization [43]. $\phi_\gamma(\cdot)$ and $\phi_\beta(\cdot)$ are two learned affine transformations, which are implemented as $1 \times 1$ convolutional filters in our setting, to convert the discrete representation $z_q$ to the scaling and bias values. Note that, the output of $\phi_\gamma(\cdot)$ and $\phi_\beta(\cdot)$ hold the same resolution to the current activation $F^{i-1}$, which injects spatial variances into the discrete feature.

In fact, this spatially conditional normalization is derived from the adaptive instance normalization (AdaIN), which has been applied for many image synthesis approaches, such as style transfer [9, 17], image-to-image translation [18, 27], and image generation [20, 41]. Compared to these methods, our spatially conditional map is a quantized map, which contains learned compact contents. As such modulation operator injects the spatially variant information, it encourages the same quantization entries to generate plausible and diverse results on different locations.

**Multichannel representation.** Unlike existing quantizers that map an image into a single-channel index map, here we convert it into a multichannel index map with the shared codebook to further improve the image quality, similar to [47]. This is inspired by the conventional GAN setting, where $1 \times 1 \times n_c$ random vector is able to generate photorealistic images with reasonable structure [13, 20, 21], showing that values across the channels contain abundant information. In practice, we first subdivide the encoded continuous feature $\hat{z} \in \mathbb{R}^{h \times w \times n_z}$ along the channel dimension into multiple chunks, *i.e.* $\hat{z} = \{\hat{z}^{(1)}, \cdots, \hat{z}^{(c)}\}$, $\hat{z}^{(c)} \in \mathbb{R}^{h \times w \times n_z/c}$. Each of these chunks is then quantized based on the equation 2 to the closest codevectors in the codebook.

For $256 \times 256$ images, we downsample it by a fixed factor of 16 to $16 \times 16$ features. Unless otherwise noted, in this work we employ $c = 4$ parallel chunks, and our equivalent sequence representation is $16 \times 16 \times 4$, which is 192x times smaller than the original image. Note that, in such way the codevector dimensionality in the codebook will be reduced to $n_q = n_z/c$. Due to this reduced dimension of each codevector, the total computational cost is similar to that of the single-channel

representation with fully dimensionality. In fact, our codebook $\mathcal{Z} \in \mathbb{R}^{K \times n_q}$ has a smaller size with the number of dimensions to be $n_q = n_z/c = 256/4 = 64$.

While a higher resolution representation, *e.g.* $32 \times 32 \times 1$ or $64 \times 64 \times 1$, can also improve the reconstruction quality as in [30, 32, 45], the computational cost will be expensive for the second stage due to the longer sequence $s = h \times w$ (see Sec. 3.2 for details). In contrast, our multichannel representation maintains a smaller sequence length, *i.e.* $16 \times 16$, although each sequence token now is recomposed by four pieces, each of which corresponds to one quantization index. With such an exponential combination capacity of $K^c$ for each spatial grid feature $\hat{z}_{ij}$, the multichannel representation power becomes much larger than the $K$ entries in the original codebook.

### 3.2 Modeling Prior Distribution

While a decoder can invert the discrete image embeddings $z_q$ to produce images $x$, we need to tame a model to estimate the underlying prior distribution over the discrete space to enable image generation. Since our main goal is to improve the codebook learning stage, we directly employ the existing setup for the second stage, including the conventional autoregressive model and the latest masked generative image transformer (MaskGIT) [2].

**Autoregressive token generation.** After embedding an image $x$ into an index sequence $s = \{s_1, \cdots, s_{h \times w}\}$ of the codebook entries $z_q$, the image generation can be naturally formulated as an autoregressive next-symbol predication problem. In particular, we maximize the likelihood function:

$$p(s) = \prod_i p(s_i|s_{<i}), \tag{5}$$

where $p(s_i|s_{<i})$ is the probability of having the next symbol $s_i$ given all the previous symbols $s_{<i}$. Then, the training objective for the second stage is equal to minimize the negative log-likelihood of the whole sequence, *i.e.* $\mathcal{L} = \mathbb{E}_{x \sim p(x)}[-\log p(s)]$.

There are many choices of autoregressive generation models, such as CNN-based PixelCNN [38] and transformer-based GPT [3]. While they can sequentially produce diverse and reasonable results based on the previously generated results, the sampling process is very slow.

**Masked token generation.** To produce the index sequence efficiently, we follow the latest MaskGIT to simultaneously predict all tokens in parallel. Inspired by BERT [5], the model aims to estimate the distribution of all indices based on the visible indices in a fixed length:

$$p(s) = \prod_i p(s_i|s_{\bar{m}}), \tag{6}$$

where $s_{\bar{m}}$ denotes conditional representation with partially visible codebook indices. In practice, the masked map $s_{\bar{m}}$, with random mask ratios during training, is fed into a multi-layer bidirectional transformer to estimate the possible distribution of each masked indices, where the negative log-likelilood objective as above is also calculated between the ground-truth one-hot index and predicted index. At inference time, the model runs fixed steps and predicts all tokens simultaneously in parallel based on the previous produced visible tokens at each iteration. Then, the most confident tokens are selected out and the remaining masked tokens will be predicted in the next step condition on previous predicted tokens. We refer readers to MaskGIT [2] for more details.

**Multichannel token generation.** Since our MoVQ embeds an image into a multichannel sequence $s = \{s_1, \cdots, s_{h \times w}\}$, where $s_i \in \{0, \cdots, |\mathcal{Z}| - 1\}^c$, for each position, we inversely concatenate chunks along the channel as one token for the input, and predict $c$ indices for the output. In practice, we independently mask each index for various channels and positions. Compared to MaskGIT, the predicated indices in our model condition on three directional dependencies, *i.e.* visible indices from different positions and channels, resulting in larger diversity due to the combination capability.

Table 1: Quantitative reconstruction results on the validation splits of ImageNet [33] (50,000 images) and FFHQ [20] (10,000 images). * denotes the model we trained with the publicly available code. "Num $\mathcal{Z}$" is the number of codevectors in the codebook.

| Model | Dataset | Latent Size | Num $\mathcal{Z}$ | PSNR ↑ | SSIM ↑ | LPIPS ↓ | rFID ↓ |
|---|---|---|---|---|---|---|---|
| VQGAN [11] | | 16×16 | 1024 | 22.24 | 0.6641 | 0.1175 | 4.42 |
| ViT-VQGAN [45] | | 32×32 | 8192 | - | - | - | 3.13 |
| RQ-VAE [24] | FFHQ | 8×8×4 | 2048 | 22.99 | 0.6700 | 0.1302 | 7.04 |
| RQ-VAE [24]* | | 16×16×4 | 2048 | 24.53 | 0.7602 | 0.0895 | 3.88 |
| **Mo-VQGAN (Ours)** | | 16×16×4 | 1024 | **26.72** | **0.8212** | **0.0585** | **2.26** |
| VQGAN [11] | | 16×16 | 1024 | 19.47 | 0.5214 | 0.1950 | 6.25 |
| VQGAN [11] | | 16×16 | 16384 | 19.93 | 0.5424 | 0.1766 | 3.64 |
| ViT-VQGAN [45] | ImageNet | 32×32 | 8192 | - | - | - | 1.28 |
| RQ-VAE [24] | | 8×8×16 | 16384 | - | - | - | 1.83 |
| **Mo-VQGAN (Ours)** | | 16×16×4 | 1024 | **22.42** | **0.6731** | **0.1132** | **1.12** |

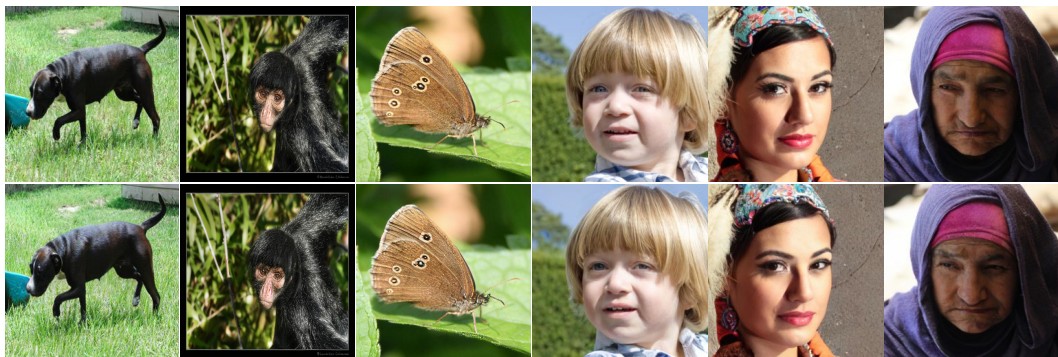

Figure 4: Top: original $256 \times 256 \times 3$ images, bottom: reconstructed images from our MoVQ with a $16 \times 16 \times 4$ latent representation in a discrete space. Zoom in to see the details.

## 4 Experiments

### 4.1 Experimental Details

**Datasets.** To evaluate the proposed method, we instantiated MoVQ on both unconditional and class-conditional image generation tasks, with FFHQ [20] and ImageNet [33] respectively. The training and validation setting followed the default one of our baseline model VQGAN [11]. We trained all the models on $256 \times 256$ images.

**Evaluation metrics.** For image generation, we used the two most common evaluation metrics Fréchet Inception Distance (**FID**) [15] and Inception Score (**IS**) [34] to evaluate the quality and diversity between generated images and ground truth images. Since we focused on the first stage, we also evaluated the quality between reconstructed images and original images. Except for the rFID score, we additionally reported the traditional patch-level image quality metrics, including peak signal-to-noise ratio (PSNR), structure similarity index (SSIM), and the latest learned feature-level LPIPS [46] for the paired reconstructed images and ground truth images.

### 4.2 Image Quantization

**Network structures and implementation details.** For each dataset, we only trained a single scale quantizer with a codebook $\mathcal{Z} \in \mathbb{R}^{1024 \times 64}$, *i.e.* 1024 codevectors each with 64 dimensions, on $256 \times 256$ images for all experiments. Our encoder-decoder pipeline is built upon the original VQGAN[1], except the only difference that we replaced the original Group Normalization with the

---

[1]https://github.com/CompVis/taming-transformers

Table 2: FID comparison for unconditional image generation on FFHQ [20] with $256 \times 256$ resolution. "# Params" is the model size and VQ-based models hold an encoder-decoder model and a prior model, respectively. "# steps" denotes the number of steps run for generating a sample.

| Model | # Params | # Steps | FID ↓ |
|---|---|---|---|
| BigGAN [1] | 164M | 1 | 12.4 |
| StyleGAN2 [21] | 30M | 1 | 3.8 |
| VQGAN [11] (w/ top-$k$ sampling) | 72.1M + 801M | 256 | 11.4 |
| ImageBART [10] | - | - | 9.57 |
| RQVAE [24] | 100M + 355M | 256 | 10.38 |
| ViT-VQGAN [45] | 599M + 1697M | 1024 | 5.3 |
| **Mo-VQGAN (Ours)-auto** | 82.7M + 307M | 1024 | 8.52 |
| **Mo-VQGAN (Ours)-mask** | 82.7M + 307M | 8 | 8.78 |

proposed spatially conditional normalization layer. We applied the new normalization layer in the first three blocks of the decoder. Following the default setting in VQGAN, images are always downsampled by a fixed factor of 16, *i.e.* from $256 \times 256 \times 3$ to a grid of tokens with the size of $16 \times 16 \times 4$. We set hyper-parameters following the baseline VQGAN work, and we trained all models with a batch size of 48 across 4 Tesla V100 GPUs with 40 epochs for this stage.

We compare MoVQ with the state-of-the-art methods for image reconstruction in Table 1. All instantiations of our model outperform the state-of-the-art methods under the same compression ratio (192x). This includes the concurrent works ViT-VQGAN [45] and RQ-VAE [24], which utilize higher resolution representation and recursively residual representation, respectively. Without bells and whistles, though we use a much smaller number of parameters (82.7M), similar to VQGAN (72.1M), MoVQ outperformers ViT-VQGAN [45], which employs a larger transformer model (599M) on higher resolution representation ($32 \times 32$) for the first stage. The concurrent RQ-VAE work [24] also represents an image with multiple channels by recursively calculating the residual information between quantized vectors and their continuous ones, which requires much more embedding times. Furthermore, the entries in RQ-VAE are *not* equally important because the residual representation highlights the codevectors in the first round, leaving the other codevectors to capture the small residual information. In contrast, our codevectors share the same significance in all channels, resulting in larger representation capability. More importantly, our model dramatically improves the reconstructed images quality on all metrics, suggesting the reconstructed images are closer to the original inputs, which contributes to more downstream image interpolation tasks.

Note that, our learned codebook contains only 1024 codevectors with 64 dimensionality, but interestingly outperforms other methods using larger codebook sizes. This suggests that a better quantizer can improve the codebook usage, and it is not necessary to greedily increase the codebook size.

The qualitative results are visualized in Figs. 3 and 4. MoVQ achieves impressive results under various conditions. In Fig. 3, we compare our MoVQ with the baseline model VQGAN [11] and the concurrent model RQ-VAE [24]. VQGAN holds repeated artifacts on the similar semantic patches, such as the grass and trees. RQ-VAE improves the visual appearance, but exhibits systematic artifacts with lossy information. Our MoVQ shows no such artifacts, providing much more realistic details.

## 4.3 Image Generation

**Network structures and implementation details.** After embedding images as a sequence, a transformer is implemented to estimate the underlying prior distribution in the second stage. Here, we used the same configuration for all models: 24 layers, 16 attention heads, 1024 embedding dimensions and 4096 hidden dimensions. The network architecture is build upon the VQGAN baseline, except that we predicted a $16 \times 16 \times 4 \times 1024$ tensor in the final layer, where $16 \times 16 \times 4$ is the number of predicted indexes, and 1024 is the number of entries in our learned codebook. Here, we set the image generation in two scenarios: (1) an "**auto**" scenario, in which the training and inference are totally similar with the VQGAN baseline, and (2) a "**mask**" scenario, in which the training and inference

Table 3: Quantitative comparison for class-conditional image generation on ImageNet [33].

| Model | # Params | # Steps | FID ↓ | IS ↑ | Prec ↑ | Rec ↑ |
|---|---|---|---|---|---|---|
| BigGAN-deep [1] | 160M | 1 | 6.95 | 198.2 | 0.87 | 0.28 |
| DCTransformer [25] | 738M | > 1024 | 36.51 | - | 0.36 | 0.67 |
| Improved DDPM [26] | 280M | 250 | 12.26 | - | 0.70 | 0.62 |
| VQ-VAE-2 [31] | 13.5B | 5120 | 31.11 | ∼45 | 0.36 | 0.57 |
| ADM [6] | 554M | 250 | 12.94 | 101.0 | 0.69 | 0.63 |
| VQGAN [11] | 1.4B | 256 | 15.78 | 78.3 | - | - |
| RQ-VAE [24] | 3.8B | 1024 | 7.55 | 134.0 | - | - |
| MaskGIT [2] | 227M | 8 | 6.18 | 182.1 | 0.80 | 0.51 |
| VIT-VQGAN [45] (w/ rejection sampling) | 2.2B | 1024 | 4.17 | 175.1 | - | - |
| **Mo-VQGAN (Ours)-auto** | 389M | 1024 | 7.13 | 138.3 | 0.75 | 0.57 |
| **Mo-VQGAN (Ours)-mask** | 389M | 12 | 7.22 | 130.1 | 0.72 | 0.55 |

is inspired by MaskGIT [2][2]. Unless otherwise noted, we default samples tokens as in MaskGIT for much faster sampling. The hyper-parameters follow the defaults setting as in VQGAN, and we trained all models with a batch size of 64 across 4 Tesla V100 GPUs with 200 epochs.

The unconditional generation results are compared in Table 2. While the original autoregressive sampling in our baseline VQGAN [11] provides tiny better generation results, the sampling time is much more expensive. Following MaskGIT, we do not set any special sampling strategies such as top-$k$ and top-$p$ sampling heuristics. We produce 60,000 samples at the inference time. Our model outperforms most VQ-based methods, with smaller parameters and faster inference time. While our model performs worse than the concurrent work ViT-VQGAN, they use a much larger model (5 times than ours) in the second stage with a longer training, which is *not* a directly fair comparison. A few generated samples are visualized in Figures 1 (see the Appendix for more). MoVQ generates photorealistic high-fidelity images, with a large diversity.

We also compare the proposed MoVQ with state-of-the-art models for class-conditional image generation on ImageNet $256 \times 256$. Our model generates 50 samples for each category, 50,000 samples in total, for the quantitative evaluation. As shown in Table 3, the proposed MoVQ significantly improve the performance over the baseline model VQGAN [11]. Compared with the concurrent work RQ-VAE [24], which also employs a multi-channel representation, our performance is competitive with them, even without the specially designed RQ-transformer for further predicting multichannel indexes. As MaskGIT and VIT-VQGAN use more GPUs for longer training, it is *not* a directly fair comparison with our MoVQ. Figures 5 show a few generated samples. Note that, the proposed MoVQ appears to be able to generate high frequency details, such as the eyes of the bird.

### 4.4  Discussion

We run a few ablations to analyze the proposed MoVQ. The results are summarized in Fig. 6.

**Architecture.**  Fig. 6 (a) reports MoVQ with various decoder settings. Compared with the baseline model VQGAN ($\mathbb{A}$), although we used the same settings for the network architecture, the multichannel representation ($\mathbb{B}$) naturally leads to a significant improvement by decomposing and recomposing the features in channel. For the spatially conditional normalization ($\mathbb{C}$), we explored three most commonly used embedding functions as the initial feature $F^0$, including positional embedding [12, 40], learned constant [20], and Fourier features [36, 19]. For positional embedding, we implemented sine and cosine functions of different frequencies on different locations and channels. The learned constant is implemented as learned parameters with the same size as the embedded discrete feature. However, the improvement from both methods is limited. We believe a key reason is the lack of special identity information for the reconstruction task, while they can provide high frequency signals. In contrast, each Fourier feature is calculated from the corresponding discrete feature, which contains the identity feature, as well as the high-frequency signals, resulting in a significant improvement.

---

[2]https://github.com/CompVis/taming-transformers

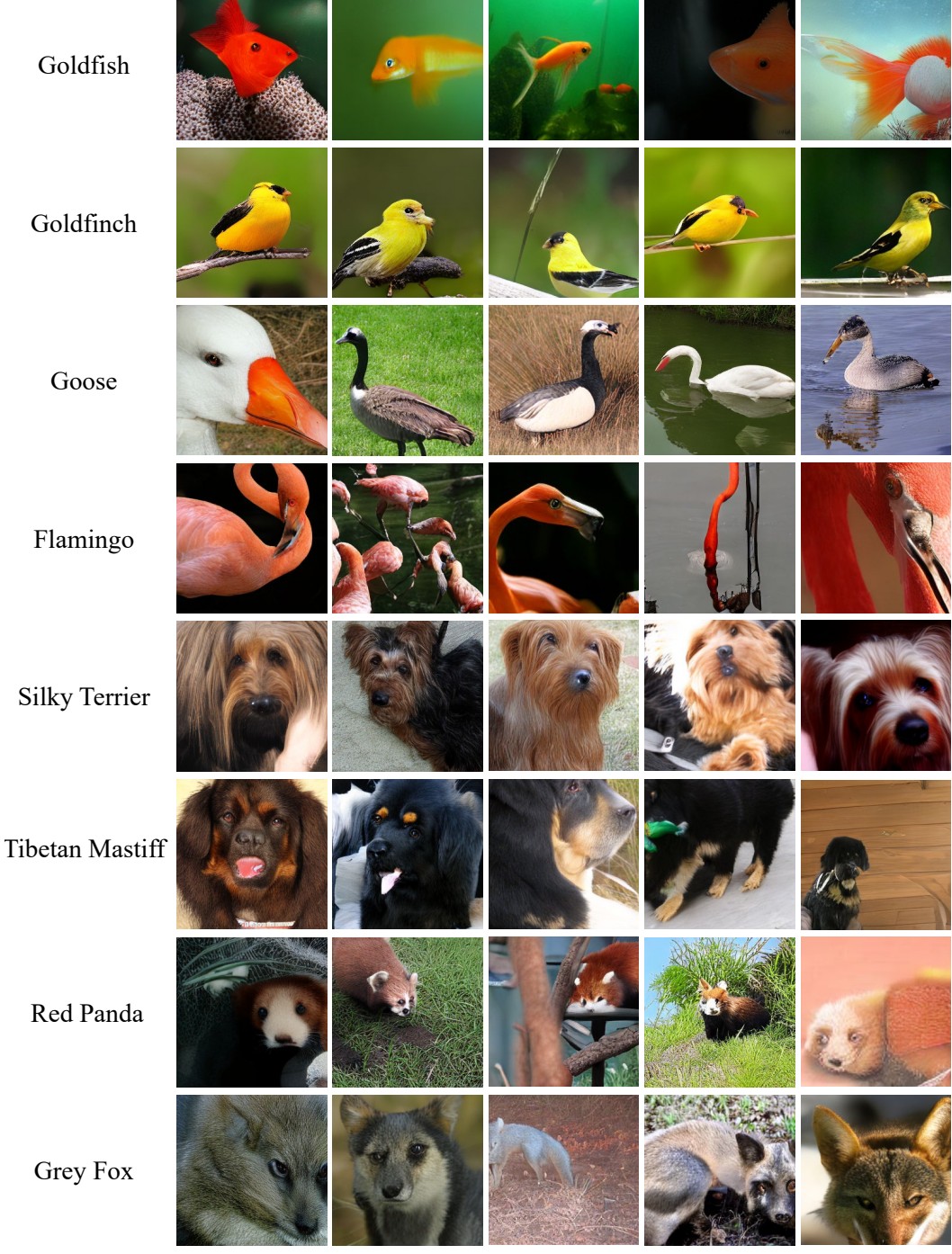

Figure 5: Generated $256 \times 256$ images by our MoVQ for class-conditional generation on ImageNet.

**Codebook size.** Vanilla VQGAN shows a larger codebook leads to a much better representation. However, this is *not* true in our experimental results. In Fig. 6 (b) we compare the results with different codebook sizes. As shown in the figure, the difference between the 1024 codebook entries and 16384 entries in our setting is negligible. This suggests that once a better quantizer is designed, it is sufficient to represent vast amounts of images using a small codebook, which makes the model easier to train. Besides, we found a low codebook usage in current VQ models, while our design can somehow improve the codebook usage. We plan to investigate this issue in future work.

| | Methods | rFID | FID |
|---|---|---|---|
| $\mathbb{A}$ | Baseline VQGAN | 4.42 | 11.4 |
| $\mathbb{B}$ | + multichannel x4 | 3.78 | 10.6 |
| | w/ sinusiods | 3.52 | 9.17 |
| $\mathbb{C}$ | w/ learned constants | 3.48 | 8.86 |
| | w/ Fourier features | **2.26** | **8.78** |

| Methods | Num $\mathcal{Z}$ | rFID |
|---|---|---|
| VQGAN | 1024 | 6.25 |
| VQGAN | 16384 | 3.64 |
| MoVQ | 1024 | 1.12 |
| MoVQ | 16384 | **1.05** |

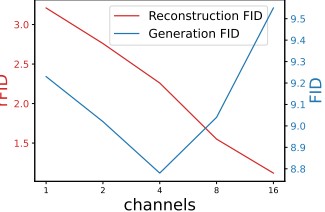

(a) Initial spatial features $F^0$ for the spatially conditional normalization on FFHQ.

(b) Different numbers of entries in the codebook on ImageNet.

(c) Reconstruction and generation FID of representation with different channels on FFHQ.

Figure 6: Ablations on MoVQ for different network architectures, codebook sizes and latent maps.

**Latent size.**   Vector quantization is a compression system, and we exploit the influence of different compress ratios by using different latent sizes. In order *not* to dramatically increase the computational cost, we mainly compare the performance of the latent representation with different numbers of channels in Fig. 6 (c). The reconstruction performance is naturally benefits from more channels due to the lower compression ratio. However, we note that not all configurations automatically benefit from more channels on image generation, as we need to predict more information in the second stage. Thus, we select a trade-off between the reconstruction and the generation. It is worth to note that this is an initial step toward combining spatial vectors in the likelihood model with the channel information as in GAN for the generation task.

## 5   Conclusion

In this work, we introduced MoVQ, a new model that is simple, efficient yet effective for generating diverse and plausible images using a better quantizer with a powerful transformer to estimating the prior. Our encoder and decoder architectures are kept simple as in the baseline framework VQGAN, except that we incorporate a spatially conditional normalization and multichannel latent maps into the VQGAN method. Experimental results show that MoVQ significantly outperforms the state-of-the-art VQ models on image modeling, yet without increasing computational cost. With a better quantizer, we show that the fidelity of our unconditional samples and class-conditional samples are better than the existing methods.

In the future, we would like to explore the semantic meaning of each entry in the learned codebook. As our model not only generates high fidelity image as the state-of-the-art GAN, but also provides much better reconstructed images, we are excited about the future of VQ-based models and plan to apply them to more image inversion, interpolation and translation tasks.

**Limitation.**   Although our MoVQ dramatically improves the image representation quality than the state-of-the-art under the same compression ratio, the model sometimes generates images with a high-frequency appearance, while the structure information is missing. We believe this is partially due to the multichannel representation we employed in our model. Therefore, a better generation model for modeling multichannel indexes needs to be further studied.

**Acknowledgements:**   This research was supported by Monash FIT research Grant.

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
