# MoVQ: Modulating Quantized Vectors for High-Fidelity Image Generation

## A  Discussion on Masked Image Reconstruction

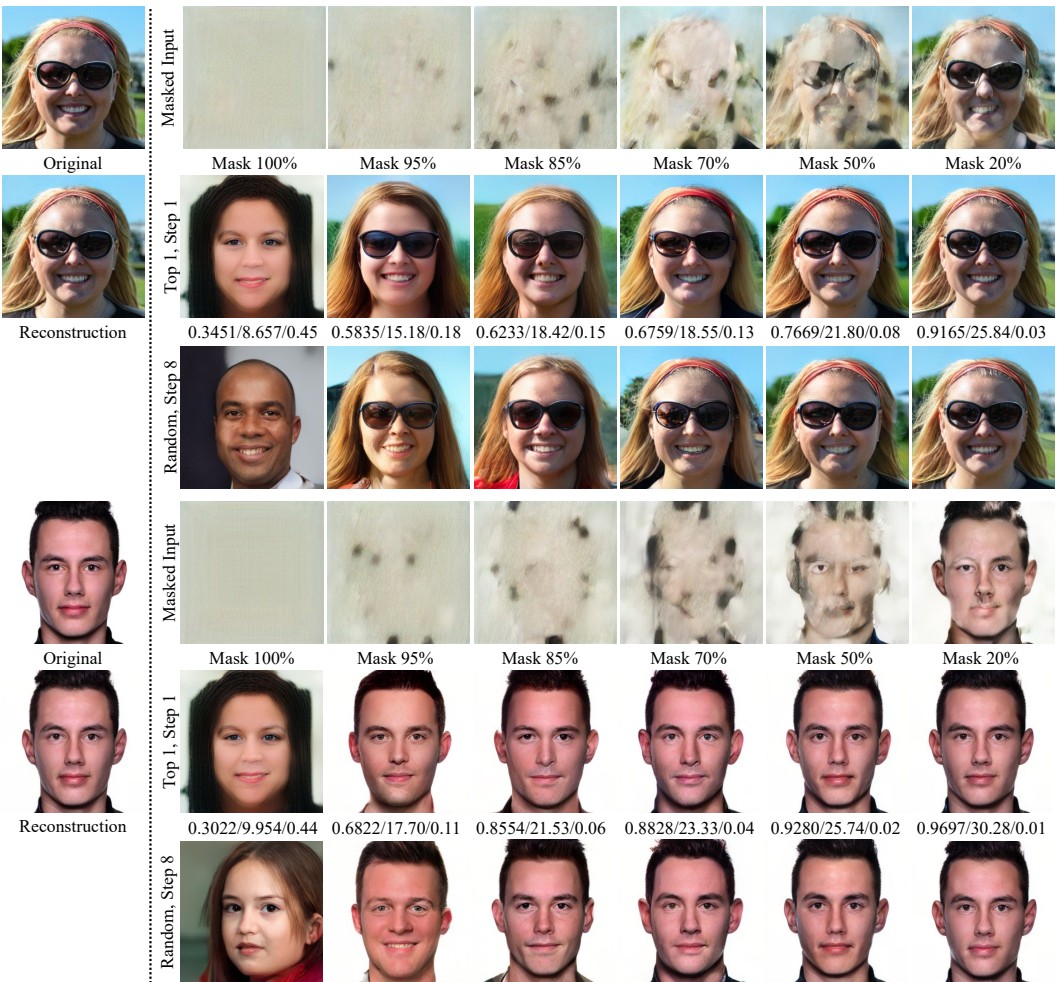

Figure A.1: Reconstruction samples for masked inputs. The first column is original images and the corresponding VQ reconstructed images in first stage. In other columns, we randomly mask some tokens (first row), and we sample the invisible tokens based on the visible tokens for the second stage. Here, we show top-1 results in 1 step (second row), and random results in 8 steps (third row), respectively. The scores are SSIMs, PSNR and LPIPS that calculate the similarity between the random results and the determinate results.

We first show the qualitative results for masked image reconstruction in the second stage. Here, a binary mask $M$ is randomly sampled based on a ratio $r$ (from 100% to 20%), and the corresponding tokens will be masked out for the zero values. Then, a bidirectional transformer is applied to predict invisible tokens conditioned on remaining visible tokens. Results are shown in Fig. A.1 and discussed in detail next.

The top-1 reconstruction results naturally benefit from more visible tokens, due to one time sampling in this case. Interestingly, our model with 95% masked tokens (*i.e.*, 12 tokens are visible among 256 tokens in each channel) is able to generate pluralistic images in only one step by selecting the top 1 token. More importantly, the corresponding results reflect identity attributes of original unmasked inputs. This suggests that the identity information is decided by only a very small portion of the tokens, especially in our multichannel representation. Therefore, following MaskGIT [**?** ],

the stochastic values are gradually reduced in the next random samples to promote a better trade-off between quality and diversity.

When the tokens are totally masked (*i.e.*, 100% mask ratio), the model generates plausible and diversity results by randomly sampling tokens in multiple steps. Note that, the top-1 results for all masked regions are a defined neutral face, which reflects the median images learned from the datasets. We also observe that the random results gradually close to the top-1 results along with more visible tokens, suggesting a lower uncertainty for more definitely visible tokens.

# B    Additional Image Generation Results

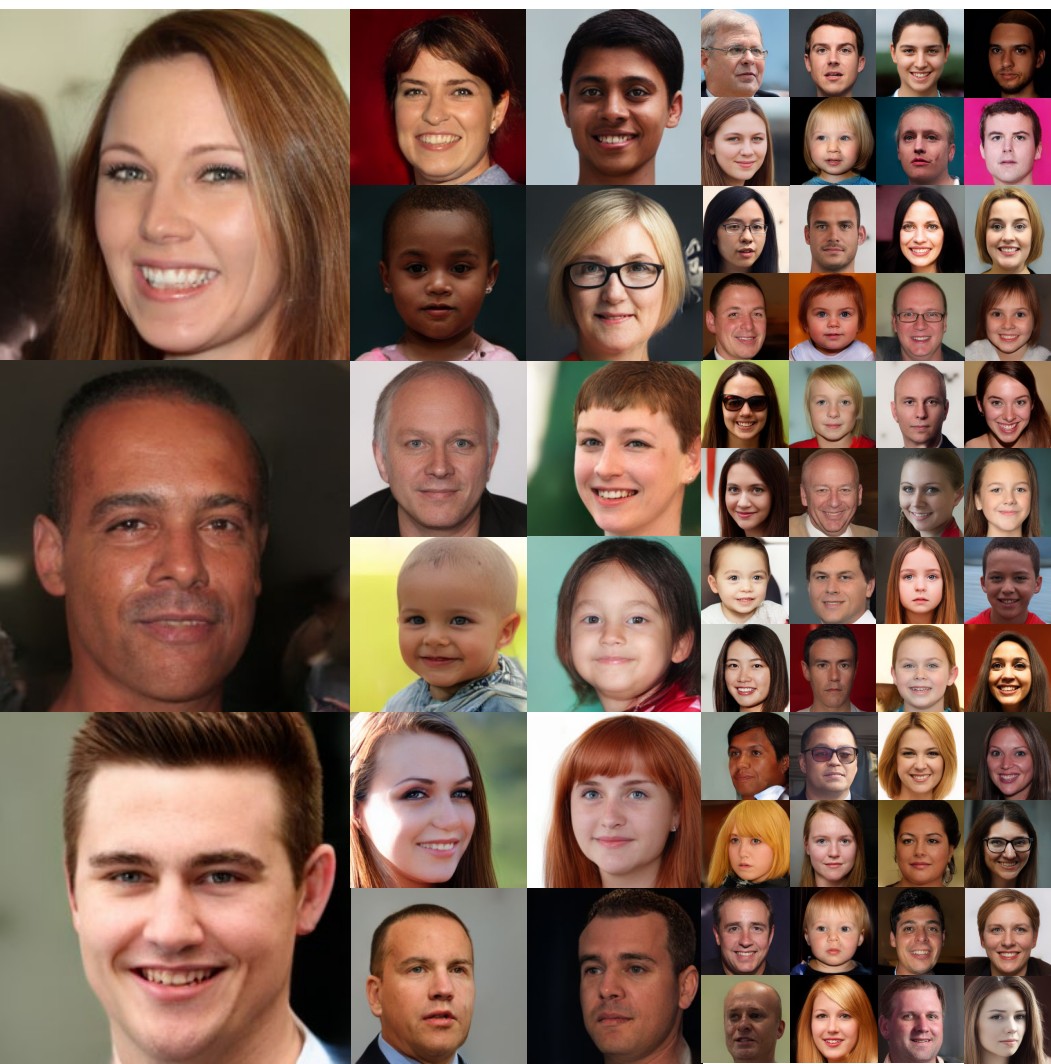

Figure B.2: More qualitative results of unconditional image generation on FFHQ dataset. To demonstrate our model is able to generate diversity results with high-fidelity, here we show various results from our first 100 samples. This is an extension of Fig.**??**

**Unconditional Image Generation.**    In Fig. B.2, we further show more samples for unconditional image generation task on FFHQ dataset. Here, we arrange the results from the first 100 samples, without additional sampling. As we can see, our model provides competitive results with the state-of-the-art GAN-based StyleGAN [**? ?** ] with large diversity and high fidelity. It is worthy to note that, our samples seem to hold much cleaner background than StyleGAN.

**Class-conditional Image Generation.** Additionally, we show more samples for class-conditional image generation on ImageNet dataset in Figs. B.4 and B.3. This is an extension of Fig. **??**. Here, all samples are conditioned on the corresponding class label with 12-step sampling. As we can see, our MoVQ generates high fidelity images with large diversity for various categories. Note that, the generated background is also clear with large diversity, such as the grass for "Red Panda".

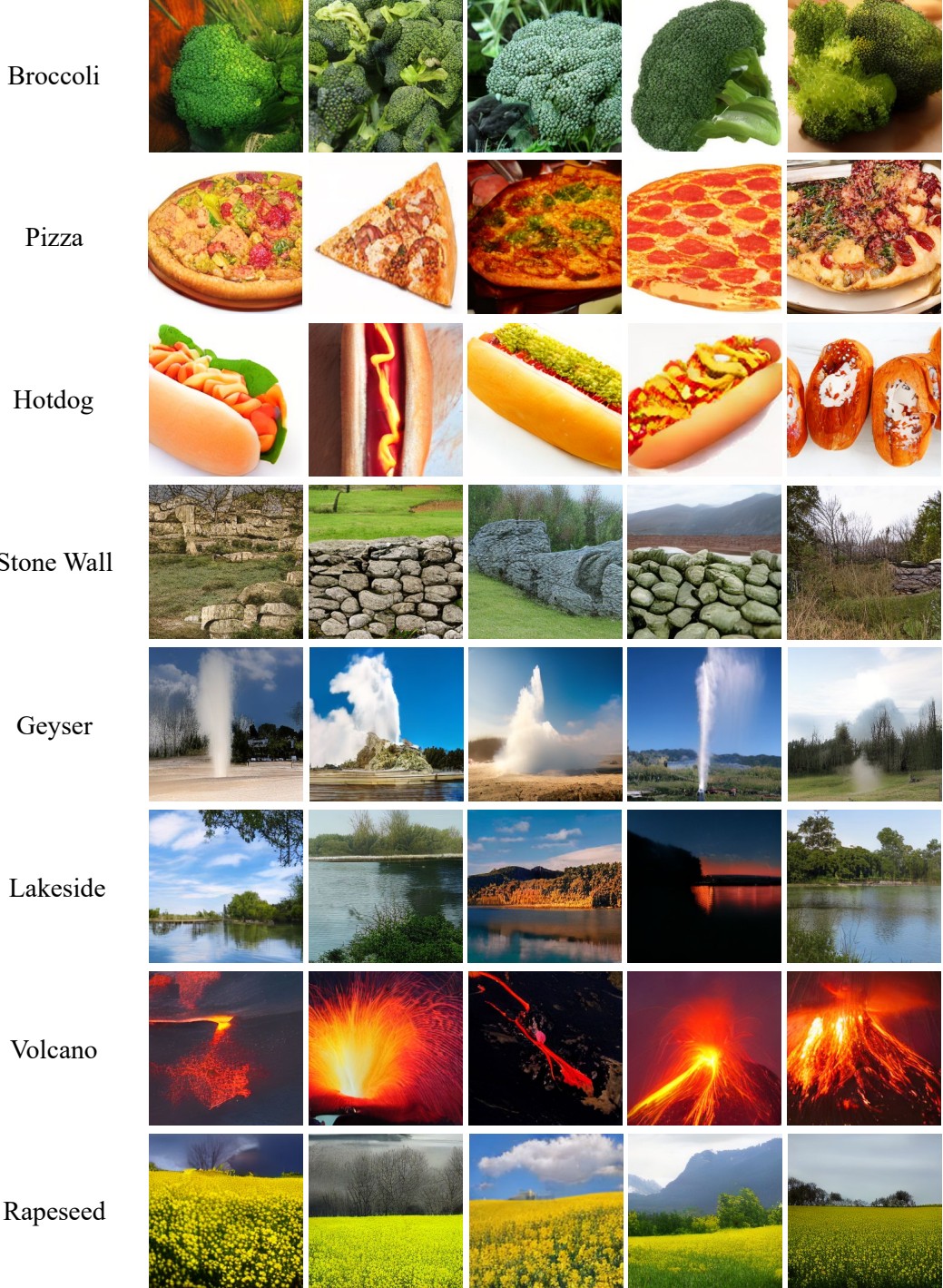

Figure B.3: More qualitative results of class-conditional image generation on ImageNet dataset. Here, we mainly show the results for foods and natural scenes. This is an extension of Fig. **??**.

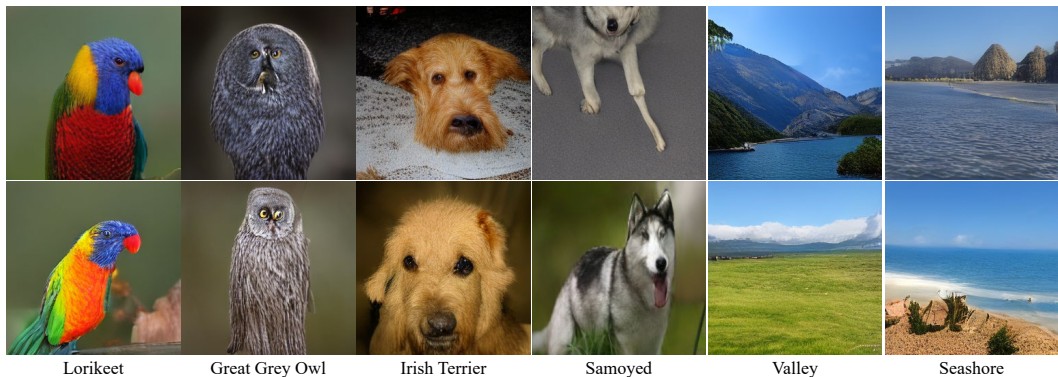

| Lorikeet | Great Grey Owl | Irish Terrier | Samoyed | Valley | Seashore |

Figure B.4: More qualitative results of class-conditional image generation on ImageNet dataset. This is an extension of Fig. **??**.