# OpenReview forum: "MoVQ: Modulating Quantized Vectors for High-Fidelity Image Generation"
_NeurIPS.cc/2022/Conference — NeurIPS 2022 Accept_

### Official Review · Reviewer_hVkj · 2022-07-07

**Rating:** 4
**Confidence:** 5
**Soundness:** 2 fair
**Presentation:** 2 fair
**Contribution:** 2 fair

**Summary:**

The paper discusses a method for generating, reconstructing images using quantized representations. The key difference wrt prior work is that they modulate the quantized representations, that is, they propose to use AdaIN-like modulation of the quantized features. They claim that without this, the results are often repetitive. A further smaller contribution is to use several channels of quantized features, ie when the image is encoded they split features into 4 blocks along the channel dimension and quantize them with the same dictionary.

They show better reconstruction results and comparable or worse results on image synthesis.

**Questions:**

Please see above

**Limitations:**

It would be great to better understand what the proposed stage-1 can do and what it cannot do. Like can it do outpainting, superresolution and other downstream applications. Reconstruction per se is a less interesting application.

**Strengths And Weaknesses:**

The paper is well written and addresses a challenging, very competitive problem. The key contribution---modulating quantized vectors---is interesting although perhaps not sufficient on its own right. Improving stage-1 training is a very important problem, which potentially can improve many applications, including image synthesis, in- and out-painting, text-to-image, video synthesis. The paper, however, needs to be improved to be able to claim that their contribution lead to these improvements.

**Motivation**: The paper is motivated by the claim that quantization results into repetitive structures. I wonder if the authors could support this statement somehow, since it's not easily observable in images if that's the case. Furthermore, if that's one of the motivations, it should be supported by numerical experiments, especially to show that the contributions improve the situation. It's not totally clear how do to this, but a method based on image autocorrelation might do the trick. The only support for this claim in the paper is fig 3. However, I cannot say it shows any repetitive content, besides, this image *can* be repetitive. Finally, even if there is repetitive content, is it due to the exactly the same tokens (a visualization of token indices will help here), or because the decoder collapses. Currently, the support behind the claim is not sufficient. Modulation can "just" improve expressivity of the the tokens, allowing the generator to have shorter tokens.

**Methodology**: from a technical standpoint modulation of the tokens is a simple extension of the original quantized schema. The proposed multi-channel quantization is interesting too, but I cannot say that both of these contributions combined together bring the paper above the bar.

**Significance**: Improving only the first stage makes the paper less interesting. Why did previous papers gain attention? I believe because they had good encoder-decoder frameworks, which offered rich latent spaces. These latent spaces can be used to solve a variety of generative tasks, such as reconstruction, generation, completion, outpainting, text-to-image and even text-to-video. If the current paper improves stage-1 it can perhaps also show better results at stage-2? In terms of stage-2 the paper reports image synthesis comparisons, in which the numbers are either on par or worse than the state-of-the-art. The authors say that their model is smaller, but the model size is not the main contribution of the work. I believe, there are many ways of making the model even smaller. If you make the model larger will it be better? In the supplement they show class-to-image generation results. According to table 3, there is no improvement over MaskGIT in terms of all of the scores. MaskGIT is even a smaller model. So in terms of significance of the proposed contributions it's hard to tell whether stage-1 improvements lead to improvements in downstream tasks. The numbers show the opposite. An intuition could be that the proposed framework reconstructs images better than others, because it provides larger, more expressive latent space, at the cost of poorer structure. Somehow the authors admit this in the limitations paragraph. It would be great if the paper could prove otherwise.

---

> ### Author Response · Authors · 2022-07-31
> **Response to Reviewer hVkj**
>
> Thank you for your constructive and detailed comments. We have revised the manuscript according to the suggestions and elaborated all concerns below.
>
> - **Q1: “...improve many applications…”**
>
> Thanks for the suggestion of applying our modulating quantized vectors to many other downstream applications, which are definitely our future work. However, as the first step, we want to focus on the reconstruction and the generation, which is also a common scope for the existing works such as VQ-VAE, VQ-VAE-2, RQ-VAE, ViT-VQGAN. The reconstruction is the cornerstone to learn the compact and expressive representation for downstream tasks. As demonstrated in RQ-VAE (CVPR’22) and ViT-VQGAN (ICLR’22), a better reconstruction naturally results in better editing applications **under the same compression ratio**. In addition, **in the appendix, we show the excellent image completion results with huge masked regions (90% masked ratio)** using our proposed simple and efficient quantizer MoVQ. Our proposed module is flexible to be integrated into other VQ-based architectures and be applied for other applications, and we leave it to the research community for future explorations.
>
> - **Q2: Motivation and methodology**
>
> **a). Motivation**: In most cases, the repeated indexes result in **"jaggies" artifacts** in the original VQGAN decoder, which can be observed in many VQGAN visual results. To further elaborate, we encoded 10,000 images on ImageNet using VQGAN (16384 entries), where there are averagely 27.21% repeated indexes.  However, this repeated **"jaggies" artifact** is difficult to be measured in synthesized images for patch similarity.
>
> **b). Significance**: While the concurrent work RQ-VAE (CVPR’22, unpublished during the submission) mitigates this issue using a residual representation in a recursive way, it requires much more embedding times. Another concurrent work ViT-VQGAN (ICLR’22, unpublished during the submission) applies larger ViT model for the embedding. Compared to these work, our key motivation is to modulate the spatial information for discrete index and preserve the tokens’ special information as in SPADE [27].
>
> **c). Conciseness**. To achieve the goal, we built our model and code upon the VQGAN baseline by simply replacing the group normalization with the proposed normalization layer, along with different types of initial spatial features in the decoder.
>
> **d). Efficiency**. Our model significantly improves the reconstruction quality than the VQGAN baseline, and the reconstruction performance is even better than the concurrent works RQ-VAE and ViT-VQGAN, suggesting the reconstructed images are closer to the original inputs, which can contribute to many downstream image generation tasks.
>
> **e). Flexibility**. As the proposed normalization layer is simple, efficient and yet effective for preserving the spatial information on discrete representation, we believe this plug-in module would be useful to the community. It can be easily reproduced and integrated into other VQ frameworks.
>
> - **Q3: Significance in terms of the generation results compared to SOTA**
>
> a). Please see our response to **Reviewer D1KP Q2** regarding the generation results.
>
> b). **Our better compact representation is not due to a larger latent space**. Our model used **the same number of token in latent space ($16\times16\times4$ (ours) vs $8\times8\times16$ (RQ-VAE) vs $32\times32$ (ViT-VQGAN)) and even a smaller codebook  (1024 (ours) vs 16483 (RQ-VAE) vs 8192 (ViT-VQGAN)) than the concurrent work RQ-VAE and ViT-VQGAN**, and yet achieved much better reconstruction quality. Besides, without a special design for the stage-2, our model achieved better performance than RQ-VAE (a larger model) on image generation.
>
> - **Q4: “Improving only the stage-1 makes the paper less interesting….”**
>
> The overall VQ-based image synthesis quality depends on both stage-1 quantizer and the stage-2 probability model. Therefore, some works (VQ-VAE-2, RQ-VAE) focus on the stage-1, and some works (VQ-DDM, MaskGiT)  focus on the stage-2, and others (VQ-GAN, ImageBART, ViT-VQGAN) address both. In this paper we focus on improving the stage-1 quantizer of the VQ-based image synthesis pipeline. We show that the proposed module leads to better reconstruction quality in stage-1 (Figures 3 and 4, and Table 1) under the same compression ratio (RQ-VAE and ViT-VQGAN), which leads to a better generation quality in stage-2 (Tables 2 and 3) than the baseline VQGAN and the concurrent work RQ-VAE under the similar training setting.

---

### Official Review · Reviewer_D1KP · 2022-07-11

**Rating:** 5
**Confidence:** 4
**Soundness:** 4 excellent
**Presentation:** 3 good
**Contribution:** 2 fair

**Summary:**

The paper presents a new VQ-based image synthesis method. Based on MaskGIT, the paper proposes spatially conditional normalization and uses multichannel representation to improve the reconstructed image quality of the tokenization stage. The proposed spatially conditional normalization modulates the quantized vectors for a better reconstruction performance by inserting spatially variant information to the VQ decoder. The multichannel representation subdivides the encoded continuous latent along the channel dimension into multiple chunks and quantizes them with a shared codebook, which further improves the reconstruction performance by increasing the latent size.  For the generation stage, the paper modifies MaskGIT to sample the multichannel latent. Experimental results on two benchmark datasets show the proposed image synthesis method is efficient and effective for generating diverse and high-quality images.

**Questions:**

1.Section 4.3 says that MaskGIT and VIT-VQGAN performs better on ImageNet because they “use more GPUs for longer training”. I’m interested in the difference of training budget. Can the proposed Mo-VQGAN surpass MaskGIT given the same training budget?
2.It would be better to conduct more ablation experiments on the generation performance. Which initialization of the spatially conditional normalization is better for the generation? Why is the spatially conditional normalization only used in the first three blocks of the VQ decoder?


**Limitations:**

The paper discusses an interesting limitation of the proposed method. The model sometimes generates images with a high-frequency appearance without the structure information, which may be attributed to the generation of multichannel representation. I think the choice of mask scheduling function in MaskGIT may be not optimal for the multichannel representation. Maybe the multi-channels of the same location should be masked and generated together, and the sampling can rely on the probability product of the multi-channels.

**Strengths And Weaknesses:**

Strengths:
+ The paper is technically sound.
+ The paper is well structured.
+ The citations are extensive.

Weaknesses:
- The novelty is somewhat limited. The idea to improve the reconstruction ability of the tokenization stage is original and interesting. However, the proposed method heavily relies on existing techniques such as multichannel representation and MaskGIT.
- The performance on image synthesis is not good enough. As shown in Table 3, the proposed Mo-VQGAN performs worse than MaskGIT for class-conditional image generation on ImageNet in terms of complexity and quality metrics. It weakens the contribution since the proposed Mo-VQGAN is based on MaskGIT.
- The evaluation is somewhat limited. The main contribution of the paper is incorporate the spatially conditional normalization to modulate the quantized vectors. Figure 6(a) has shown that spatially conditional normalization can improve the reconstruction performance (rFID), especially for Fourier features. However, improving the reconstruction performance (rFID) does not necessarily improve the generation performance (FID), as shown in Figure 6(c). In my opinion, the experimental evaluation should report FID with or without the proposed spatially conditional normalization.

---

> ### Author Response · Authors · 2022-07-31
> **Response to Reviewer D1KP**
>
> Thanks for your constructive comments and detailed suggestions.
>
> - **Q1:”The novelty is somewhat limited…”**
>
> a). Please see our response to **Reviewer hVkj Q2**.
>
> b). Regarding MaskGIT, first of all, we did not claim the adoption of MaskGIT as a contribution. **MaskGIT is only used for the faster sampling and our performance gain is not due to MaskGIT**. We believe it is important to facilitate researchers in this area to do fair comparisons efficiently, for which generating examples faster is critical, since the previous way of autoregressive sampling is too slow. To demonstrate our improvements mainly come from our special design in the stage-1, we additionally report the quantitative results with autoregressive sampling like VQGAN in revision. The autoregressive sampling achieves a slight better FID score (8.52 vs 8.78 on FFHQ, and 7.13 vs 7.22 on ImageNet), compared to the MaskGIT sampling. However, to sample 60,000 examples, the autoregressive sampling takes 10 days with a batch size of 12 in 4 V100 GPU, while MaskGIT takes about 3 hours.
>
> - **Q2: “The performance on image synthesis is not good enough…”**
>
> a). It is hard to give a fair comparison with MaskGIT (CVPR’22) and VIT-VQGAN (ICLR’22). MaskGIT is trained with 16 CloudTPUv4 with batch size of 256 for 300 epochs training in 4 days, and VIT-VQGAN is trained with 128 ColudTPUv4 with batch size of 256 for 500,000 steps training in 36 hours. We can only access 4 shared Tesla V100 GPUs, which cannot handle such a large batch size with such long training steps. However, our proposed spatially conditional normalization is a plug-in module that can be easily integrated to those architectures to further improve their performance.
>
> b). During the submission, these excellent works VIT-VQGAN and MaskGIT are not officially published, and the corresponding training codes are not publically released. Thus, we also cannot retrain them according to our setting for a fair comparison. We listed them as baselines simply to report the latest concurrent works.
>
> c). As claimed in Line 206-243, **our network architecture, code and hype-parameters are built upon the VQGAN baseline (CVPR’21)**. For MaskGIT, we just adopted their parallel sampling strategy. Compared to the VQGAN baseline, our model significantly improves the performance on image reconstruction and generation. Even compared with the concurrent work RQ-VAE (CVPR’22, unpublished during the submission), our model achieves better generation quality using smaller codebook size with same numbers of tokens. Our model significantly improves the reconstruction quality, which can be very helpful for other downstream tasks such as image inpainting, interpolation and editing.
>
> - **Q3: “The evaluation is somewhat limited…”**
>
> The scores in Figure 6(c) are for **different channels corresponding to different numbers of tokens**. It is naturally challenging to predict more tokens for the probability models in the stage-2. However, **under the same compression ratio, i.e. the same number of tokens**, a better rFID score indicates a better compact representation, leading to a better generation score. This is verified in the table below. Besides, the concurrent works VIT-VQGAN (ICLR’22) and RQ-VAE (CVPR’22) also claimed that a better quantizer is definitely helpful for the generation task. Moreover, thanks to the suggestion, we added the ablation FID results with or without the proposed spatially conditional normalization for generation (see the table below as well as the updated Fig.6(a)), which clearly demonstrates the superiority of our proposed module.
>
> | | Methods | rFID | FID|
> | -- | -- | -- | -- |
> | $\mathbb{A}$ | Baseline VQGAN  | 4.42 | 11.4 |
> | $\mathbb{B}$ | + multichannel x4  | 3.78 | 10.6 |
> | | w/ sinusiods | 3.52 | 9.17 |
> | $\mathbb{C}$ | learned constants |  3.48 |  8.86 |
> | | w/ Fourier features | 2.26 | 8.78 |
>
> - **Q4: Regarding questions**
>
> Most of the questions have been addressed above. As for why the proposed normalization is only used in the first 3 blocks of the decoder, it is simply because features in the first 3 blocks hold the same resolution in VQGAN baseline. We directly apply the conditional discrete map ($16\times16\times4$)  into these layers. If applying the normalization layer in other blocks, we will need to learn multiscale representations or upsample the discrete map to different resolutions, resulting in a complex predication for stage-2.
>
> - **Q5: Regarding limitations**
>
> Thanks for this insightful comment. We indeed have some initial experiments on simultaneously masking and sampling multichannels tokens, but the observed performance is slightly worse. We guess this may because some token in some special channels might be significant, which needs to be sampled first. For instance, we might first generate an eye token and then predict its color. As the stage-2 is not the focus of this paper, we leave it for our future investigation.

---

### Official Review · Reviewer_GwN2 · 2022-07-11

**Rating:** 7
**Confidence:** 4
**Soundness:** 3 good
**Presentation:** 3 good
**Contribution:** 3 good

**Summary:**

The author's introduce a new VQ-GAN model with three improvements, spatial normalization of the quantized vectors, MaskGIT for quicker autoregressive reconstruction, and multichannel feature quantization.  They show better reconstruction performance with similar code size to other models and methods.

**Questions:**

Other papers have found that generalized normalization functions can learn as powerful representations as much deeper traditional neural network architectures without normalization.  Do the author's think this may be part of the power of the addition of normalization as they state it?  Or is it something specific to normalization before the MASKGit?

**Ethics Review Area:**

["I don’t know"]

**Limitations:**

The authors address stock concerns about implications but I believe that larger implications are raised by the image of the child in Figure 4 (4th column).  The child's face has changed significantly from the original, but unlike traditional encoding techniques, does not betray any indication to the downstream user that the image has lost information or is in any sense "uncertain".  Traditional artifacts in simpler compression techniques may look bad to the eye but they at least faithfully convey to the user when information has been lost.  I think not enough concern is paid in this paper and in this literature to technologies that produce confident and clear images that are not what was captured and encoded on the other side, and that may fool the end user otherwise.

**Strengths And Weaknesses:**

This is obviously of interest to the NeurIPS community and the results are impressive.  The implication that normalization the quantized code vectors adds substantial improvement is interesting, and definitely opens up interesting areas to follow up on.  The use of MASKGit seems to me to be not a large contribution of this work, and while interesting, can be downplayed a bit in comparison to the normalization and use of multichannel quantization.

The weaknesses are that the paper is not always careful in its comparison to other methods (figure 3 doesn't show the code size or latent size in comparison between several different methods so it is fairly difficult to compare across methods).  More careful use of common tools from compression literature (on rate-distortion) would help clarify some of the comparisons across methods.

---

> ### Author Response · Authors · 2022-07-31
> **Response to Reviewer GwN2**
>
> Thank you for your constructive comments and recognition of our MoVQ model.
> - **Q1: “...not always careful in its comparison to other methods…”**
>
> Thanks for the suggestion. We have reported the codebook size and the latent size in Table 1 for a fair comparison on all models. The same configuration is used in Figure 3. We have added this detail into the revised version. Compared to the latest state-of-the-art RQVAE [24] (CVPR’22, unpublished during this submission), our model significantly improves the image quality in the first stage under the same compression ratio, while using much smaller codebook size (1024 entries vs 16384 entries).
>
> - **Q2: “...generalized normalization functions…”**
>
> Thanks for this insight comment and interesting discussion. Different from the traditional normalization, which is for better statistical learning, our proposed spatially conditional normalization is designed to improve the current VQ based generation framework, which often embeds similar neighboring patches into the same quantization index and leads to repeated artifact patterns in the generated images. Specifically, the proposed normalization layers provide different scale and shift values according to the different quantized values, as well as adapt to different spatial locations. Currently, this spatial normalization is specific to the discrete token index map. For the conventional network architecture (such as MaskGIT), we need to design the specific conditional map as the spatially-variant input.
>
> - **Q3: Regarding limitations**
>
> Thanks for the insightful comment. The observation to the child image in Fig.4 is interesting. Zooming into this particular example, we do agree that the constructed image does have some perceived identity or attribute difference from the original image. This might be due to the FFHQ dataset contains much more young people (34,654 (age:20-40)) than children (9,873 (age:0-10)). On the other hand, we did provide detailed quantitative comparisons in all popular metrics including PSNR (pixel level), SSIM (patch level), LPIPS (feature level), rFID (dataset level) in Table 1. Compared to the existing VQ-based methods, the proposed method has significantly improved the reconstruction quality under the same compression ratio. As for traditional compression techniques, they can indeed be faithful to each individual image, but are unable to utilize the power dataset priors like the data-driven approaches. Thus, their compression ratios are limited, and the compressed images may not be photorealistic.

---

### Author Response · Authors · 2022-08-10
**Looking forward to your reply**

Dear reviewers,

We first thank you again for your valuable comments and detailed suggestions. In the previous replies, we have tried our best to address your questions and revised the manuscript based on the suggestions.

We are looking forward to your reply to our responses, and we are open to any discussions to improve this work.

Best wishes!

---

### Meta-Review · Area_Chair_xp5o · 2022-08-27

**Recommendation:** Accept
**Confidence:** Less certain

**Metareview:**

The three reviewers had significantly diverging final opinions (strong accept, borderline accept and weak reject). The authors addressed many of the concerns in their rebuttal. I read the paper carefully, and I agree with the concerns from one reviewer about why the improvements in stage-1 do not lead to significant improvements in stage-2. I think this concern needs to be properly addressed, because otherwise it is unclear what the benefit of this approach would be for real applications. While previous work has shown that improved stage-1 performance leads to improved stage-2 performance, why was it not replicated in this situation? I also found the analysis of why the spatially conditioned normalization improves reconstruction to be lacking. If the "jagged" structures are addressed by this work, then understanding why with simple examples would have shed more insight into the technical contribution. However, in summary, I think this paper is slightly above the acceptance bar, and addressing the above concerns is recommended for the final version.

**Award:**

No

---

### Decision · Program_Chairs · 2022-09-14

Accept